# Eccentric force and electromyogram comparison between the eccentric phase of the Nordic hamstring exercise and the razor hamstring curl

**Yuta Murakami[1], Satoru Nishida[2]\*, Kaziki Kasahara[1], Riku Yoshida[3], Ryo Hayakawa[4], Masatoshi Nakamura[5]**

1 Institute for Human Movement and Medical Sciences, Niigata University of Health and Welfare, Niigata, Japan, 2 Faculty of Sports and Health Science, Ryutsu Keizai University, Ibaraki, Japan, 3 Department of Rehabilitation, Medical Corporation, Maniwa Orthopedic Clinic, Niigata, Japan, 4 Department of Rehabilitation, Medical Corporation Sansuikai, Kitachiba Orthopedic, Makuhari Clinic, Chiba, Japan, 5 Faculty of Rehabilitation Sciences, Nishi Kyushu University, Saga, Japan

\* satoru.nishida5521@gmail.com

**Data Availability Statement:** All relevant data are within the paper and its Supporting Information files.

## Abstract

### Purpose

Nordic hamstring exercise (NHE) and razor hamstring curl (RHC) are usually performed to train hamstring eccentric contraction strength. However, it is unclear whether there are differences in the intensity of the two methods and the amount of loading on each muscle. Therefore, this study was conducted using peak eccentric force and each muscle surface electromyogram (s-EMG) to provide useful information to decide whether NHE or RHC should be prescribed for training and rehabilitation.

### Methods

s-EMG electrodes were placed in the medial gastrocnemius, lateral gastrocnemius, biceps femoris, semitendinosus, gluteus maximus, and erector spinae of the dominant leg of the fifteen healthy male university students with exercise habits. Maximum voluntary isometric contractions of 3 seconds were performed on each muscle followed by NHE and RHC in random order. The outcome variables included peak eccentric force and s-EMG of each muscle calculated by means amplitude during the NHE and RHC.

### Results

Peak eccentric force was significantly higher in RHC than in NHE ($p = 0.001$, $r = 0.73$). However, NHE was significantly higher in s-EMG of semitendinosus ($p = 0.04$, $r = -0.52$) than RHC. However, there were no significant differences in EMG of the medial gastrocnemius ($p = 0.202$, $r = -0.34$), lateral gastrocnemius ($p = 0.496$, $r = 0.18$), biceps femoris ($p = 0.061$, $r = -0.48$), gluteus maximus ($p = 0.112$, $r = -0.41$), erector spinae ($p = 0.45$, $r = 0.20$) between NHE and RHC.

**Funding:** This work was supported by the JSPS KAKENHI (Grant Number JP20K19435), but the funders had no role in study design, data collection and analysis, decision to publish, or preparation of the manuscript.

**Competing interests:** The authors have declared that no competing interests exist.

## Conclusions

For NHE and RHC, the peak eccentric force exerted during the exercise was significantly higher for RHC, and the s-EMG of semitendinosus was significantly higher for NHE.

## Introduction

The hamstring strain injury (HSI) is common and accounts for 12%–15% of all sports injuries [1]. A previous study [2] has showed that eccentric contraction (ECC), which occurs during the late swing phase of sprinting, was linked to HSI. The hamstring ECC strength is critical for preventing HSI [2–5].

Nordic hamstring exercise (NHE) is a training in which the patient gradually leans forward from a kneeling posture with the ankle fixed and the knee joint flexed 90˚ [6, 7]. NHE is beneficial for preventing HSI [8] because hamstring ECC muscle strength [9–11], and fascicle length [8, 9] increase. Mjølsnes et al. [10] investigated the effects of a 10-week NHE training program on muscle strength in male soccer players. The results showed that there was an 11% increase in ECC hamstring torque at 60˚/s and a 7% increase in isometric hamstring strength at 90˚, 60˚, and 30˚ of knee flexion. Additionally, Vianna et al. [9] studied the NHE practices twice a week for 8 weeks with a gradually increased training load in female professional soccer players. The results showed an increase in the fascicle length of the long head of the biceps femoris (BF) of approximately 6%. Additionally, a study found that performing NHE before and after training reduced the incidence of HSI in professional soccer players by 92% compared with the previous season [11]. Furthermore, a recent meta-analysis [12] found that training programs that include NHE can reduce HSI by up to 51% compared to training programs without NHE. Therefore, NHE is effective for HSI prevention.

Similarly, razor hamstring curl (RHC) is a training for the hamstrings [13–17]. Although similar to NHE in some aspects, RHC is a method in which the knee and hip joints are fully flexed, and the hip and knee are simultaneously extended closely to the supine position (full hip and knee extension) [16]. Previous studies [13, 14] have examined surface electromyogram (s-EMG) exerted on the BF, medial hamstrings, gluteus maximus (GM), and vastus medialis, and have reported that high s-EMG is exerted especially on the BF and medial hamstrings. Although NHE and RHC are ECC exercises for the hamstrings, the training outcomes of these two approaches are different. Similarly, to compare the training effects of NHE and RHC, Pollard et al. [16], conducted a study in three groups; NHE with body weighted (NHE$_{bodyweight}$), NHE with weighted (NHE$_{weighted}$) and RHC with weighted (RHC$_{weighted}$) in a 6-week training intervention. The results showed a significant increase in fascicle length of the long head of the BF only in NHE$_{weighted}$ mice. Similarly, van den Tillaar et al. [17] has shown that NHE was significantly higher than RHC in the s-EMG of semitendinosus (ST), semimembranosus, and BF. Therefore, the hamstring s-EMG exerted by NHE is higher than that by RHC. A previous study showed that s-EMG of the GM, Erector Spinae (ES), and gastrocnemius were exerted during NHE [18]. However, no previous studies have compared the NHE and RHC in s-EMG of muscles other than the hamstrings. Recently, instruments have been developed to measure the peak eccentric force during NHE and RHC. The peak eccentric force is considered an indicator of hamstring ECC strength, such as used to predict the risk of HSI [19, 20]. Measuring peak eccentric force in addition to s-EMG can provide insights into the load on each muscle during NHE and RHC. However, no previous studies have compared peak eccentric force during NHE and RHC. Comparing each muscle's s-EMG and peak eccentric force during NHE

and RHC, and clarifying whether there are differences in exercise intensity and the amount of load on each muscle, provide useful information to decide the appropriate method for training and rehabilitation. In a previous study [17], NHE was reported to have higher hamstring s-EMG than RHC. Additionally, NHE calls for the patient to lean forward while maintaining an intermediate hip joint and trunk position, which is predicted to be more challenging for upper body control than RHC. Therefore, the s-EMG of the hamstrings, GM, and ES may be greater in the NHE than in the RHC. In a previous study by Pollard et al. [16], no significant differences were found in peak eccentric force during NHE and RHC in the baseline.

This study compared peak eccentric force and each muscles' s-EMG during the NHE and RHC. The s-EMG of the hamstrings, GM, and ES may be greater in the NHE than in the RHC, and we hypothesized that there would be no significant difference in peak eccentric force during NHE and RHC.

## Method

### Participants

This study included the fifteen healthy male university students (age of 20.9 ± 0.9 years, height of 171.1 ± 5.1 cm, and weight of 68.2 ± 9.0 kg) with habitually performed resistance training including that for the knee flexors 2–3 times a week. This study excluded the persons who had experienced his, spondylolysis, or ligamentous injury within the previous 12 months. The sample size was calculated using G*Power 3.1 software (Heinrich Heine University, Düsseldorf, Germany) using larget effect size, alpha error of 0.05 and power of 0.80 required for the t-test and resulted 15 subjects. Participants were invited to apply from March 1, 2022. Information identifying individual participants was accessible to us during and after data collection. To properly protect participant information, personally identifiable information was anonymized.

This study was conducted with the approval of the Ethical Review Committee of Fukuoka University (Ethics approval number: #21-04-M1). Study procedures and potential risks were explained to the participants, and each participant provided a written informed consent before participation in the study.

### Experimental procedures

Participants were asked to visit the laboratory twice. The first visit was scheduled 7 to 10 days before the measurement date. During the first visit, the NHE and RHC movements were explained and the participants practiced each movement. At the measurement date, participants performed warm-ups consisting of 10 deadlifts using a weight of approximately 10 kg and 10 split jumps (5 on each leg). Then, s-EMG electrodes were applied to the dominant legs' medial gastrocnemius (MG), Lateral Gastrocnemius (LG), BF, ST, GM, and ES. Moreover, three seconds of maximum voluntary isometric contractions (MVIC) was performed on each muscle. After that, strength testing for NHE and RHC was conducted randomly.

### Nordic hamstring exercise and razor hamstring curl strength testing

Fig 1 shows the setting of the NHE and RHC. As described previously [6, 7], each participant was placed in a kneeling position on a custom-made NHE device, and each ankle was secured over the lateral ankle by an ankle brace attached to a load cell. The force against the ankle brace in the vertical direction was measured by the load cell connected to a PowerLab16/35 (AD Instruments, Bella Vista, Australia), synchronously transferred the force data to a personal computer (VersaPro; NEC, Tokyo, Japan) at 1000 Hz [6]. The NHE [19, 20] was performed in which the participants were instructed to lean forward gradually at the slowest

(A)

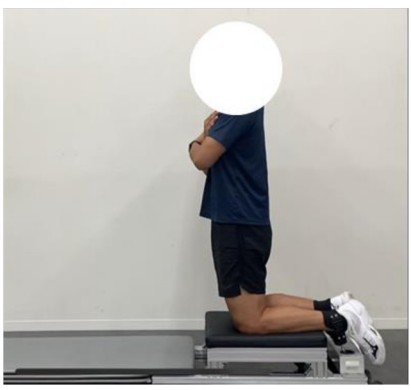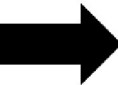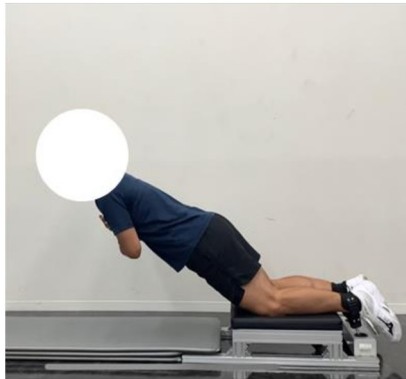

(B)

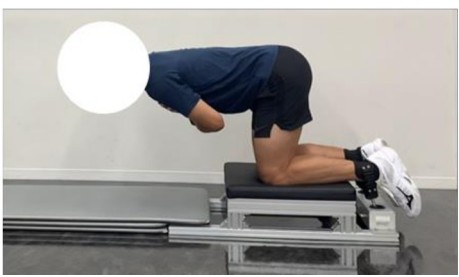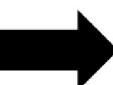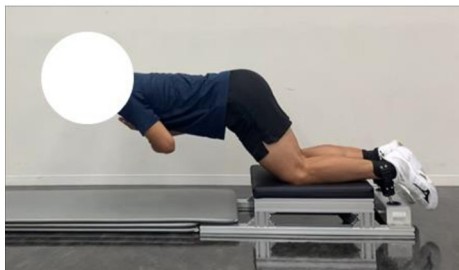

**Fig 1.** Experimental set-up for Nordic hamstring exercise (A) and Razor Hamstring Curl (B).

possible speed from a kneeling posture with the knee joint flexed 90˚ to a prone position with the arms crossed at the chest and the hip joint fully extended. In contrast, for RHC, participants were instructed to change the kneeling position, with their buttocks immediately superior to their ankles and their knees, and hips fully flexed [16]. Participants were then instructed to simultaneously extend both the hip and knee, maintaining a consistent distance between their head and the mat, to reach a near-prone position (full hip and knee extension). Each movement was practiced 2–3 times before the measurement. At the time of the measurement, the examiner provided verbal feedback on whether the participant was performing the exercise appropriately. Both NHE and RHC were performed twice in the measurement. In addition, the order of measurement was randomized among the participants, and the maximum value of the peak eccentric force to body weight ratio (N/kg) of the dominant leg was used for further analysis.

## Surface electromyography

The position of the s-EMG electrode attachment was based on Surface Electromyography for the Non-Invasive Assessment of Muscles [21]. After skin treatment of each site with alcohol cotton before measurement, the following surfaces were covered with electrodes: MG, site of

significant muscle bulging; LG, proximal 1/3 of the line connecting the fibular head and heel; BF, the midpoint between the sciatic tubercle and lateral epicondyle of tibia; ST, midpoint between the sciatic tubercle and medial epicondyle of tibia; GM, the midpoint between the sacrum and greater trochanter; ES, 2–3 cm lateral to the third lumbar vertebra. The location of the muscle was confirmed by palpation of landmarks and isometric contraction. s-EMG derivation was performed by the bipolar method using surface electrodes, with the distance of 20 mm between the center electrodes [17], and recorded on a computer via a preamplifier (FA-DL-720-140: Four Assist, Japan) and AD converter (Power Lab: AD instrument) to a personal computer (DESKTOP-CRVN2SU, MouseComputer Co., Ltd., Japan). The band-pass filter was set from 20 to 500 Hz, and full-wave rectified s-EMG waveforms were analysed by the software (LabChart 8.1.14: AD instrument, Australia). Full-wave rectified s-EMG waveforms smoothed using a root mean square filter with a moving window of 50 milliseconds, the analysis interval was defined as the start of the operation until the end of the operation of the NHE, RHC, and MVIC. The mean amplitudes obtained during NHE, RHC, and MVIC were used as s-EMG values. s-EMG at MVIC was used for normalization of the s-EMG at NHE and RHC (%MVIC). For the MVIC measurement method, MG and LG were measured in the sitting position with knee 0˚ flexion and plantar 0˚ flexion using a multi-purpose muscle function evaluation and training device (BIODEX system 3.0: BIODEX, Inc., Shirley, NY, USA). BF and ST were measured in the supine position with the knee in 45˚ flexion and the hip in 0˚ extension. GM was measured in the supine position with the knee flexed 90˚ and the hip extended. ES was measured in the supine position with both hands on the head, knees of both legs flexed to 0˚, and body extended. Moreover, the larger value obtained in s-EMG measured twice each at NHE and RHC was used in the analysis.

## Statistical analysis

SPSS 28.0 J (SPSS, Japan) was used for statistical processing. According to the results of the Shapiro-Wilk test, all continuous data were normally distributed except BF and s-EMG of GM during NHE and LG, and s-EMG of GM. To compare peak eccentric force and s-EMG obtained in NHE and RHC, the Wilcoxon signed-rank test was used to compare the EMG of LG, BF, and GM, while the variable with normal distribution were compared using the paired $t$-test. The effect size, which indicates the magnitude of the difference between peak eccentric force and s-EMG obtained at NHE and RHC, was calculated based on $r$. For the variables for which a paired $t$-test was used, the following formula was used to calculate: $r = \sqrt{(t^2/[t^2+df])}$ [22]. In addition, the following formula was used for the variables using Wilcoxon's signed-rank test: $r = Z/\sqrt{n}$ [23]. In this case, $r$ = 0.1–0.3 as representing a small, 0.3–0.5 as a moderate, and $\geq$ 0.5 as a large magnitude of change [24]. A $p$-value of <0.05 indicated statistical significance.

## Results

The peak eccentric force was significantly higher in RHC than in NHE ($p$ = 0.001, $r$ = 0.73) (Fig 2). However, NHE was significantly higher in s-EMG of ST ($p$ = 0.04, $r$ = −0.52) than RHC (Table 1). In addition, there were no significant differences in s-EMG of MG ($p$ = 0.202, $r$ = −0.34), LG ($p$ = 0.496, $r$ = 0.18), BF ($p$ = 0.061, r = −0.48), GM ($p$ = 0.112, $r$ = −0.41), ES ($p$ = 0.45, $r$ = 0.20) between NHE and RHC.

## Discussion

In this study, the peak eccentric force and s-EMG were compared during the NHE and RHC. The s-EMG of each muscle was calculated using the mean amplitude. The peak eccentric force

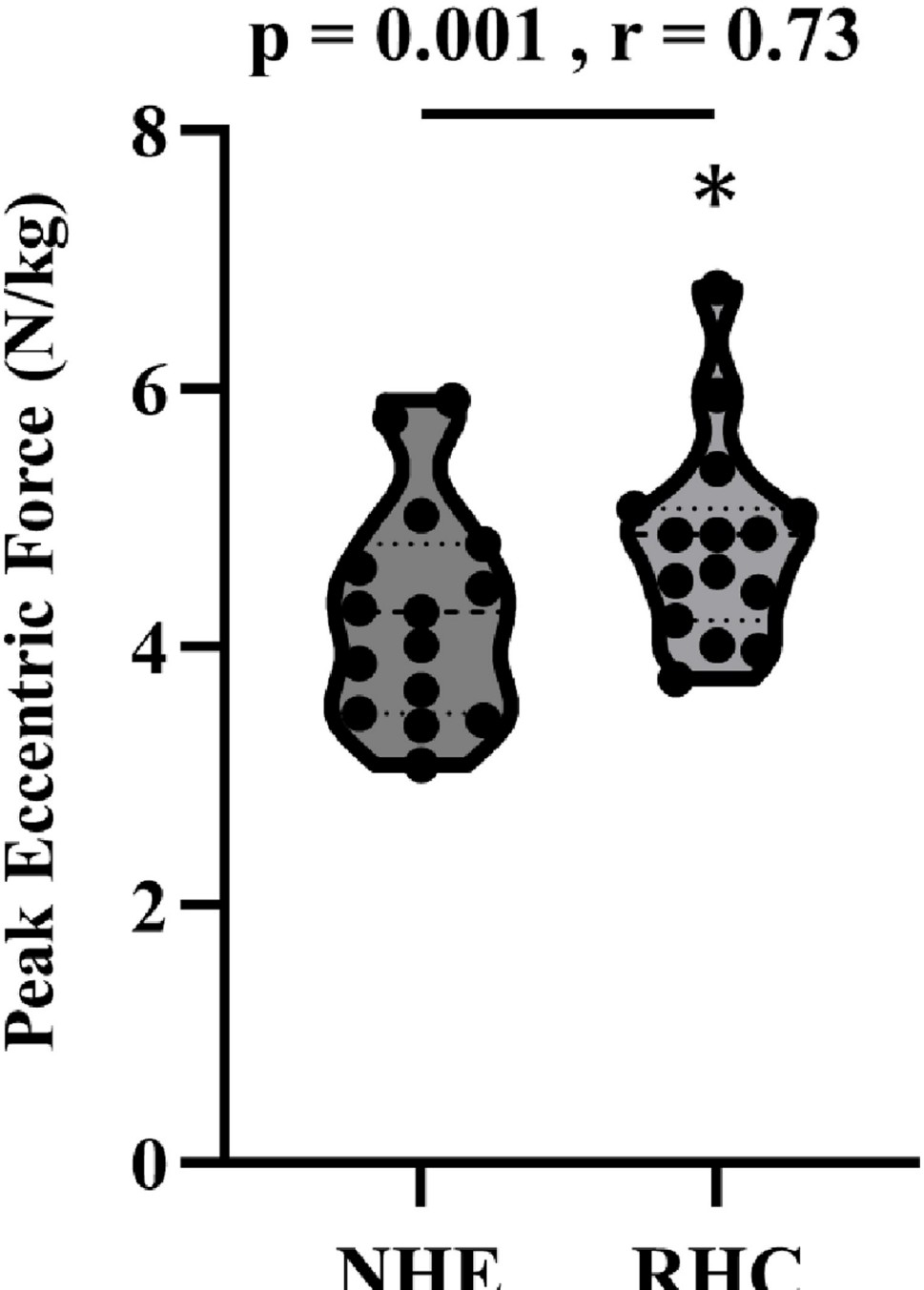

**Fig 2. Comparison of the peak eccentric force during the Nordic hamstring exercise (NHE) and razor hamstring curl (RHC).**

was significantly higher in the RHC ($p = 0.001$, $r = 0.73$). In contrast, the s-EMG of ST was significantly higher in the NHE ($p = 0.04$, $r = −0.52$). However, no significant differences in the s-EMG of BF ($p = 0.061$, $r = −0.48$), GM ($p = 0.112$, $r = −0.41$), or ES ($p = 0.45$, $r = 0.20$) were detected between NHE and RHC. Therefore, our findings refute the hypothesis that peak eccentric forces of NHE and RHC are not different. However, the results partially support the

**Table 1. Compare the mean amplitude of each muscle during the Nordic hamstring exercise (NHE) and razor hamstring curl (RHC) throughout the entire range of motion.** The original data is shown in S1 File.

| (%MVIC) | NHE | RHC | Interaction effect |
|---|---|---|---|
| Medial Gastrocnemius | 28.7±9.0 | 25.9±7.5 | p = 0.202 |
| | | | r = -0.34 |
| Lateral Gastrocnemius | 22.8±14.6 | 23.5±13.1 | p = 0.496 |
| | | | r = 0.18 |
| Biceps Femoris | 52.8±29.3 | 43.2±17.7 | p = 0.061 |
| | | | r = -0.48 |
| Semitendinosus | 49.2±13.9* | 41.7±16.9 | p = 0.04 |
| | | | r = -0.52 |
| Gluteus Maximus | 30.8±41.6 | 25.7±34.0 | p = 0.112 |
| | | | r = -0.41 |
| Erector Spinae | 37.5±9.5 | 39.7±13.4 | p = 0.45 |
| | | | r = 0.20 |

*: significant ($p < 0.05$) difference from the RHC value.

hypothesis that the s-EMG measurements of BF, ST, GM, and ES are greater during the NHE compared with the respective s-EMG values during RHC. To the best of our knowledge, this is the first study comparing peak eccentric force and s-EMG of muscles other than the hamstring between NHE and RHC.

Our results demonstrate that the peak eccentric force is significantly higher in RHC compared with NHE ($p = 0.001$, $r = 0.73$). This result may be attributed to differences in muscle length during exercise. Previous studies showed that passive tension generated by a series of elastic elements increases with longer muscle lengths [25], and passive tension contributes to exerted muscle force during an eccentric contraction [26, 27]. The RHC may result in a longer hamstring muscle length than NHE due to the emphasis on hip flexion. The longer hamstring length may cause higher passive tension and exert a higher eccentric force. Šarabon et al. [28] investigated the torque exerted during NHE with 0˚, 25˚, 50˚, and 75˚ of hip flexion; NHE with 75˚ hip flexion resulted in a larger peak torque than NHE with 0˚ hip flexion. Hegyi et al. [29] also showed that NHE at 0° hip flexion exerted more torque than NHE at a neutral (NHE at 0˚ hip flexion) hip flexion. These results indicate that performing NHE with hip flexion, i.e., the hamstring is longer, increases the eccentric force. Šarabon et al. [28] also reported that NHE with hip flexion increases the torque exerted from the hip joint. Since RHC is performed with hip flexion, the torque exerted from the hip joint may be greater than the torque while performing NHE, resulting in a significantly greater eccentric force.

Our results revealed that the s-EMG of ST was significantly higher during the NHE ($p = 0.04$, $r = −0.52$). Tillaar et al. [17] reported that performing the NHE resulted in higher BF and ST s-EMG compared to RHC. Similar to peak eccentric force, this result may be influenced by differences in the hamstring muscle length during the exercise. A narrative review by Kellis et al. [30] revealed that longer hamstring muscle length decreases EMG. Kirk et al. [31] also suggested that lengthening the muscle decreases the motor unit firing rate. The hip flexion angle is larger during RHC compared with the angle during NHE; therefore, hamstring muscle length is also longer during RHC, resulting in a relatively larger s-EMG of the STs during NHE compared with s-EMG during RHC. Tillaar et al. [17] reported that s-EMG of the BF and ST were higher during NHE compared to s-EMG during RHC and suggested that the higher s-EMG is due to the different hamstring muscle lengths. Šarabon et al. [28] and Hegyi et al. [29] also reported that NHE performed in the limb position with a smaller hamstring hip

flexion angle resulted in significantly higher s-EMG for BF and ST than s-EMG for larger hip flexion angles. These results suggest that the hamstring is lengthened by hip flexion, resulting in a smaller s-EMG. In addition, no significant differences in s-EMG of GM ($p = 0.112$, $r = -0.41$) and ES ($p = 0.45$, $r = 0.20$) were detected between NHE and RHC. We hypothesized that NHE would result in higher s-EMG values for GM and ES than RHC because NHE requires upper body control to keep the hip joint and trunk in the mid-position while performing the movement. However, our findings did not support this hypothesis. This may be due to insufficient upper body control during NHE. Moreover, our study population included healthy male college students who performed resistance training, including resistance training for the knee flexors, 2–3 times a week. Athletes who underwent high-intensity training were not included in the study. Therefore, the s-EMG of the GM and ES involved in upper body control may be incomplete due to the overload of movement execution while controlling the upper body. In addition, the hip flexion angle during NHE was not specified in this study. Therefore, the hip flexion angle increased during NHE, which may have affected the s-EMG of GM and ES. Future studies should be conducted while defining the hip flexion angle to explore this issue.

The results of this study showed that the peak eccentric force exerted during the RHC was higher than the peak eccentric force exerted during NHE, suggesting that RHC may be preferable for exerting high force throughout the lower extremities, especially in the knee and hip joints. Previous studies demonstrated that ECC with longer muscle length is more effective in increasing muscle strength than ECC with shorter muscle length [32]. Still, muscle damage is also higher in RHC [33, 34]. Since RHC is performed from the hip flexed position, the degree of hamstring muscle damage may be higher than damage when NHE is performed. Nishida et al. [6] detected no significant relationship between peak torque during NHE and knee flexion torque measured with an isokinetic dynamometer in the supine position. Therefore, the eccentric force measured in this study may reflect the combined strength of various muscles, such as the hip, and trunk extensors, gastrocnemius, and hamstring. The factors related to the eccentric force exerted during NHE and RHC should be investigated. The s-EMG exhibited by the ST during NHE was higher than the s-EMG of the ST during RHC. Therefore, NHE may be preferable if the goal is to achieve high s-EMG in the ST. The ST is used in anterior cruciate ligament (ACL) reconstruction. Therefore, the side from which the ST tendon is harvested for ACL reconstruction will exhibit decreased knee flexion muscle strength [35–37] (measured with an isokinetic dynamometer), ST s-EMG [37], eccentric force [38], and ST activation [39] during NHE compared with the corresponding parameters on the opposite side. Therefore, NHE may be suitable as an exercise prescription to improve knee flexion strength and ST s-EMG after ACL reconstruction. In summary, when prescribing NHE and RHC in training and rehabilitation, the characteristics of each exercise should be matched with the objectives of each individual.

This study has several limitations. First, healthy male university students were recruited for this study. Thus, the results may not apply to persons with prior HSI. Second, the sample size could not be calculated based on the effect sizes of previous studies, and the sample size may have been too small. Future studies should be conducted with more participants. Third, previous studies set failure criteria for hip flexion angle and discarded NHE with hip flexion angles higher than 20˚ [40]. In this study, subjects were verbally instructed not to flex their hip joints during NHE without specifying the hip joint flexion angle. Therefore, future studies should define and examine the hip flexion angle. Fourth, both NHE and RHC were performed at the slowest possible speed. Since movement speed affects muscle activation, future research should standardize movement speed using a metronome or similar device. Fifth, although the breakpoint angle during NHE was measured using an electrogoniometer in a previous study [6],

this study did not consider the breakpoint angle. Therefore, the ability to control falling forward during NHE and RHC was not compared. A previous study [6] reported a significant correlation between peak eccentric torque and the breakpoint angle obtained by NHE. Therefore, differences in breakpoint angles may contribute to the higher peak eccentric force of RHC compared to NHE. The measurement of breakpoint angles may need to be included in future studies.

## Conclusion

Peak eccentric force was significantly higher in RHC, whereas ST s-EMG was significantly higher in NHE. RHC may be desirable if the goal is to generate high force throughout the lower extremities, including the knee joint and hip joint. On the contrary, NHE may be desirable if the goal is to generate high s-EMG at the ST.

## Supporting information

**S1 File. The data set used in this study.**
(XLSX)

## Acknowledgments

The authors gratefully acknowledge all participants involved in this study.

## Author Contributions

**Conceptualization:** Satoru Nishida, Masatoshi Nakamura.

**Data curation:** Yuta Murakami, Kaziki Kasahara, Riku Yoshida, Ryo Hayakawa.

**Formal analysis:** Yuta Murakami.

**Funding acquisition:** Satoru Nishida.

**Investigation:** Ryo Hayakawa.

**Methodology:** Satoru Nishida, Masatoshi Nakamura.

**Project administration:** Masatoshi Nakamura.

**Writing – original draft:** Yuta Murakami.

**Writing – review & editing:** Satoru Nishida, Masatoshi Nakamura.

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
