## [Decision Letter · Decision Letter 0]

7 Jul 2023

PONE-D-23-17165

Comparison of eccentric force, electromyogram in Nordic hamstring exercise and Razor hamstring curl

PLOS ONE

Dear Dr. Nishida,

Thank you for submitting your manuscript to PLOS ONE. After careful consideration, we feel that it has merit but does not fully meet PLOS ONE’s publication criteria as it currently stands. Therefore, we invite you to submit a revised version of the manuscript that addresses the points raised during the review process.

We look forward to receiving your revised manuscript.

Kind regards,

Esedullah Akaras

Academic Editor

PLOS ONE

Journal Requirements:

 "We do not have any potential conflict of interest." 

   "This work was supported by the JSPS KAKENHI (Grant Number JP20K19435), but the findings of the study are independent from the funding organisation."

Additional Editor Comments:

Our reviewers were undecided about the acceptance of the article. The article is acceptable if you can make the requested revisions. Good luck.

Reviewers' comments:

Reviewer's Responses to Questions

**Comments to the Author**

1. Is the manuscript technically sound, and do the data support the conclusions?

Reviewer #1: Partly

Reviewer #2: Partly

Reviewer #3: Yes

Reviewer #4: Yes

2. Has the statistical analysis been performed appropriately and rigorously? 

Reviewer #1: Yes

Reviewer #2: Yes

Reviewer #3: Yes

Reviewer #4: Yes

3. Have the authors made all data underlying the findings in their manuscript fully available?

Reviewer #1: Yes

Reviewer #2: Yes

Reviewer #3: Yes

Reviewer #4: Yes

4. Is the manuscript presented in an intelligible fashion and written in standard English?

Reviewer #1: Yes

Reviewer #2: No

Reviewer #3: Yes

Reviewer #4: No

5. Review Comments to the Author

Reviewer #1: Thank you for allowing me to review this manuscript. The topic is of interest to clinicians and sports specialists. However, after a careful review, the paper has some methodological flaws that impact the general conclusions.

The biggest flaw is the number of participants (n=15). It is a small number to obtain general conclusions. When I repeated the power analysis using a type I error of 0.05, type II error of 0.2 and large effect size, the sample size obtained is 27 subjects. Moreover, this large effect size has not been obtained from previous studies, as it is not indicated. Therefore, it is not possible to know if this is sufficient to obtain conclusive results.

Specific revisions:

Title: “Comparison of eccentric force, electromyogram in Nordic hamstring exercise and Razor hamstring curl” should be revised. Comparison between exercises? Between muscles? Between eccentric and concentric phases?

ABSTRACT

Methods and results should include which type of EMG outcomes you use.

“Near significant” is not significant.

METHODS

Please, reference the “Surface Electromyography for the Non-Invasive Assessment of Muscles.

I suggest explaining how you assess MVIC.

EMG signal was recorded during all movement? Did you use all the signals?

Did you clean and shave the skin before the placement of the electrodes?

When you said “maximum EMG values”, do you mean maximum amplitude? Please, revise it through the text.

The maximum amplitude of EMG was normalized too?

RESULTS

Near significantly higher means no significantly higher. I suggest correcting it through the text to not misunderstand statistics results.

Table 1: “Average electromyogram” is not an outcome. You should use for example “amplitude”.

DISCUSSION

Please, reference all your sentences, as lines 215-223.

Line 231, reference 24 is repeated.

Please, include “mean amplitude” or “maximum amplitude” to clarify EMG outcomes information.

All the information regarding EMG outcomes of BF should be corrected since it is not significant. And the same through all the text.

Reviewer #2: Abstract

Abstract is generally understandable and fluent.

Line 43-45; these expressions 'in the stretched position' and ’optimal length of the hamstrings’ seems more commented.

Introduction

Line51-53 and line 67-69; Instead of explaining how the exercises are done here, explaining in the method section may make the flow better.

In studies using surface EMG (s-EMG) while providing literature information, the expression s-EMG should be used instead of EMG.

I recommend moving Line 97-103 before Line 96. After Line 96, the hypothesis should be written and this section should be closed.

Method

Line 108-109; Participants' physical activity levels should be stated more objectively.

Line 124; warm-up should be explained.

Line 126-127; Explain how MVIC was measured (which position etc.) and reference

Line 130;

• Video recording should be taken during exercises and used for S-EMG analysis and to check that the exercises are done properly. Please write if received.

• Also, were any verbal or visual warnings given to the participants? (For them to do the exercises properly)

• The speed of exercise should be standardized as it can affect muscular activation. If not, write in the limitations.

• How angles were standardized during NHE and RHC?

• Line 133-137; A reference is required that this method is valid and reliable.

Line 152-159; Reference should be given for electrode placement.

Line 171; Is post-power analysis done?

Discussion I would recommend making the discussion section more fluent. Here it is nice to support and compare your results with current literature. However, while the study was about peak ecc force and muscular activations, the discussion was based on morphological features such as fascicle length. More relevant literature on the study subject and hypothesis can be used.

Finally, you can define the peak ecc force during NHE and RHC exercises more clearly (for which muscle(s)), which highlights your working hypothesis (which is different from the current literature).

References

It's good that about 50% of the references were published in the last five years.

Reviewer #3: Thank you very much for allowing me to review this manuscript.

Overall the manuscript is very well structured, from the theoretical framework, the objectives, the analyses performed and the results obtained are consistent throughout and well written in an easily understandable way.

In my opinion, the manuscript can be published but I would recommend the authors to modify the figures so that the pictures provided are of high quality and without background that could affect the understanding of it. They should also modify the figure accompanying the results and make it much more visual, either by using a design programme or by modifying it to violin plots, which are much more appropriate.

In this way, the manuscript would gain a lot of quality and would be much more understandable and more likely to be cited.

Reviewer #4: It is an interesting article.

However, in some parts, it is necessary to add more information.

The detail is below.

L74 In this part, you wrote too much about Pollard's experiment. Please sum up this experiment.

L85 I think writing "in addition to the hamstrings" is unnecessary.

L87 Please write the detail about this measurement. It would help if you wrote why the peak eccentric force measurement is essential.

L109 About the exercise habit, this is essential information. For example, it differs depending on whether it is a power or endurance sport. Please write the detail about it.

L133 What is the load cell sensor? Please explain this. Furthermore, write the detail of this equipment; name, company...

6. PLOS authors have the option to publish the peer review history of their article (what does this mean?). If published, this will include your full peer review and any attached files.

Reviewer #1: No

Reviewer #2: No

Reviewer #3: **Yes: **Fernando Martin-Rivera

Reviewer #4: No

---

## [Author Response · Author response to Decision Letter 0]

22 Aug 2023

Reviewer #1

Thank you very much for reviewing our manuscript and providing valuable comments and suggestions that have helped us improve our manuscript. We have responded to your comments below and revised our manuscript accordingly. 

In the revised manuscript, the changes are highlighted in yellow with red letters. 

We hope that you will find our responses and revisions, considering your comments, adequate and that the manuscript has improved. We appreciate your further review of our manuscript and hope we have an opportunity to publish it in this journal.

We appreciate you sharing your indications. First, we use the mean amplitude for further analysis instead of the maximum amplitude for the electromyogram measure. We hope you can confirm this, and we apologize for any confusion this may have caused.

1. The number of participants (n=15) is small and previous studies have not shown a large effect size.

Response: Thank you very much for your indication. To the best of our knowledge, no previous cross-sectional studies have compared eccentric force and EMG obtained with NHE and RHC to calculate effect sizes. Therefore, it was impossible to determine the effect size to be used in G*Power based on the effect sizes obtained from prior studies. Therefore, we determined the sample size using the largest effect size, an alpha error of 0.05, and a power of 0.80.The study's small participant pool will be added to the list of limitations.

Change: lines 293-295

Second, the sample size could not be calculated based on the effect sizes of previous studies, and the sample size may have been too small. Future studies should be conducted with more participants.

2. Title: “Comparison of eccentric force, electromyogram in Nordic hamstring exercise and Razor hamstring curl” should be revised. Comparison between exercises? Between muscles? Between eccentric and concentric phases?

Response: We comprised eccentric force, and electromyogram between Nordic hamstring exercise and Razor hamstring curl. Thus, we have changed the title as follows to clarify this point.

Change: lines 1-3

Eccentric force and electromyogram comparison between the eccentric phase of the Nordic hamstring exercise and the razor hamstring curl

ABSTRACT

3. Methods and results should include which type of EMG outcomes you use.

Response: Thank you very much for your indication. We added a statement indicating that the EMG was calculated using mean amplitude.

Change: lines 31-33

The outcome variables included peak eccentric force and s-EMG of each muscle calculated by means of amplitude during the NHE and RHC.

4. “Near significant” is not significant.

Response: We have changed the expression "Near significant" to not significant, as you suggested.

Change: lines 37-40

However, NHE was significantly higher in s-EMG of semitendinosus (p = 0.04, r = −0.52) than RHC. However, there were no significant differences in s-EMG of the medial gastrocnemius (p = 0.202, r = −0.34), lateral gastrocnemius (p = 0.496, r = 0.18), biceps femoris (p = 0.061, r = −0.48), gluteus maximus (p = 0.112, r = −0.41), erector spinae (p = 0.45, r = 0.20) between NHE and RHC.

METHODS

5. I suggest explaining how you assess MVIC. 

Response: Thank you very much for your indication.We added the following descriptions to the MVIC measurement method.

Change: lines 172-179 

For the MVIC measurement method, MG and LG were measured in the sitting position with knee 0° flexion and plantar 0° flexion using a multi-purpose muscle function evaluation and training device (BIODEX system 3.0: BIODEX, Inc., Shirley, NY, USA). BF and ST were measured supine with the knee in 45° flexion and the hip in 0° extension. GM was measured supine with the knee flexed 90° and the hip extended. ES was measured in the supine position with both hands on the head, knees of both legs flexed to 0°, and body extended.

6. EMG signal was recorded during all movement?

Response: s-EMG signals were recorded during Nordic hamstring exercise (NHE), Razor hamstring Curl (RHC), and MVIC operations.

7. Did you use all the signals?

Response: Thank you very much for your indication. We caluculated root-mean-square filter with a moving window of 50 milliseconds used them for further analysis. We have added the following text.

Change: lines 16-171

Full-wave rectified s-EMG waveforms smoothed using a root mean square filter with a moving window of 50 milliseconds, the analysis interval was defined as the start of the operation until the end of the operation of the NHE, RHC, and MVIC.

8. Did you clean and shave the skin before the placement of the electrodes?

Response: Thank you very much for your indication. We added a sentence about treating the skin with alcohol before each measurement.

Change: lines 154-155

After skin treatment of each site with alcohol cotton before measurement, the following surfaces were covered with electrodes :

9. When you said “maximum EMG values”, do you mean maximum amplitude? Please, revise it through the text.

Response: We appreciate you sharing your indications. First, we use the mean amplitude for further analysis instead of the maximum amplitude for the electromyogram measure. We hope you can confirm this, and we apologize for any confusion this may have caused. We have revised the text as follows.

Change: lines 179-180

 Moreover, the larger value obtained in s- EMG measured twice each at NHE and RHC was used in the analysis.

10. The maximum amplitude of EMG was normalized too?

Response: Thank you very much for your indication. In this study, the maximum amplitude of s-EMG was also normalized. However, we use the mean amplitude for further analysis instead of the maximum amplitude.

RESULTS

11. Near significantly higher means no significantly higher. I suggest correcting it through the text to not misunderstand statistics results.

Response: Thank you very much for your indication. This has also been corrected in the same way as ABSTRACT.

Change: lines 198-201

However, NHE was significantly higher in s-EMG of ST (p = 0.04, r = −0.52) than RHC (Table 1). In addition, there were no significant differences in s-EMG of MG (p = 0.202, r = −0.34), LG (p = 0.496, r = 0.18), BF (p = 0.061, r = −0.48), GM (p = 0.112, r = −0.41), ES (p = 0.45, r = 0.20) between NHE and RHC.

12. Table 1: “Average electromyogram” is not an outcome. You should use for example “amplitude”.

Response: Thank you very much for your indication. As you indicated, we will use word “amplitude”.

Change: lines 203-205

Compare the mean amplitude of each muscle during the Nordic hamstring exercise (NHE) and Razor Hamstring Curl (RHC) throughout the entire range of motion.

DISCUSSION

13. Line 231, reference 24 is repeated.

Response: Thank you very much for finding our mistake. 

Change: lines227-229

Šarabon et al.[28] investigated the torque exerted during NHE with 0°, 25°, 50°, and 75° of hip flexion; NHE with 75° hip flexion resulted in a larger peak torque than NHE with 0° hip flexion.

14. Please, include “mean amplitude” or “maximum amplitude” to clarify EMG outcomes information.

Response: Thank you very much for your indication. I will take your suggestion into consideration and add the words mean amplitude.

Change: lines209

The s-EMG of each muscle was calculated using the mean amplitude.

15. All the information regarding EMG outcomes of BF should be corrected since it is not significant. And the same through all the text.

Response: Thank you very much for your indication. We have changed the information on EMG outcomes of BF throughout the entire text.

Change: lines211-213

However, there were no significant differences in s-EMG of BF (p = 0.061, r = −0.48), GM (p = 0.112, r = −0.41), and ES (p = 0.45, r = 0.20) between NHE and RHC.

lines237-238

Our results revealed that the s-EMG of ST was significantly higher during the NHE (p = 0.04, r = −0.52).

Lines311-312

Peak eccentric force was significantly higher in RHC, whereas ST s-EMG was significantly higher in NHE.

Reviewer #2

ABSTRACT

1. Line 43-45; these expressions 'in the stretched position' and ’optimal length of the hamstrings’ seems more commented.

Response: Thank you very much for your indication. This was a leap of text to describe in the Abstract's Conclusions.

Change: lines 42-44

For NHE and RHC, the peak eccentric force exerted during the exercise was significantly higher for RHC, and the s-EMG of semitendinosus was significantly higher for NHE.

Introduction

2. Line51-53 and line 67-69; Instead of explaining how the exercises are done here, explaining in the method section may make the flow better.

Response: Thank you very much for your indication. The text on how to do the exercises will be provided in the Method section.

lines 137-142

The NHE [19, 20] was performed in which the participants were instructed to lean forward gradually at the slowest possible speed from a kneeling posture with the knee joint flexed 90° to a prone position with the arms crossed at the chest and the hip joint fully extended. In contrast, for RHC, participants were instructed to change the kneeling position, with their buttocks immediately superior to their ankles and their knees, and hips fully flexed [16].

3. In studies using surface EMG (s-EMG) while providing literature information, the expression s-EMG should be used instead of EMG.

Response: Thank you very much for your indication. For studies using surface EMG, the term be changed to "s-EMG".

4. I recommend moving Line 97-103 before Line 96. After Line 96, the hypothesis should be written and this section should be closed.

Response: Thank you very much for your indication. As you indicated, we have changed the order of the sentences.

Change: lines 88-102

Comparing each muscle s-EMG and peak eccentric force during NHE and RHC, and clarifying whether there are differences in exercise intensity and the amount of load on each muscle, provide useful information to decide the appropriate method for training and rehabilitation. In a previous study [17], NHE was reported to have higher hamstring s-EMG than RHC. Additionally, NHE calls for the patient to lean forward while maintaining an intermediate hip joint and trunk position, which is predicted to be more challenging for upper body control than RHC. Therefore, the s-EMG of the hamstrings, GM, and ES may be greater in the NHE than in the RHC. In a previous study by Pollard et al.[16], no significant differences were found in peak eccentric force during NHE and RHC in the baseline. 

This study compared peak eccentric force and each muscle s-EMG during the NHE and RHC. The s-EMG of the hamstrings, GM, and ES may be greater in the NHE than in the RHC, and we hypothesized that there would be no significant difference in peak eccentric force during NHE and RHC.

METHODS

5. Line 108-109; Participants' physical activity levels should be stated more objectively.

Response: Thank you very much for your indication. We have included details regarding the physical activity level of the participants.

Change: lines 106-108

This study included the fifteen healthy male university students (age of 20.9 ± 0.9 years, height of 171.1 ± 5.1 cm, and weight of 68.2 ± 9.0 kg) with habitually performed resistance training including that for the knee flexors 2–3 times a week.

6. Line 124; warm-up should be explained.

Response: Thank you very much for your indication. Added detailed information on the warm-up.

Change: lines 122-124

At the measurement date, participants performed warm-ups consisting of 10 deadlifts using a weight of approximately 10 kg and 10 split jumps (5 on each leg).

7. Line 130; Video recording should be taken during exercises and used for s-EMG analysis and to check that the exercises are done properly. Please write if received. Also, were any verbal or visual warnings given to the participants? (For them to do the exercises properly)

Response: Thank you very much for your indication. No videotaping was done during the exercise, but the participants were given a verbal warning to ensure they were exercising properly.

Change: lines 145-146

 At the time of the measurement, the examiner provided verbal feedback on whether the participant was performing the exercise appropriately.

8. Line 130; The speed of exercise should be standardized as it can affect muscular activation. If not, write in the limitations. 

Response: Thank you very much for your indication. In this study, both NHE and RHC were performed at the slowest possible speed. Therefore, as you mentioned, it is possible that the speed varied and affected muscle activation. The following document has been added to the limitations.

Change: lines 299-301

Fourth, both NHE and RHC were performed at the slowest possible speed. Since movement speed affects muscle activation, future research should standardize movement speed using a metronome or similar device.

9. Line 130; How angles were standardized during NHE and RHC?

Response: Thank you very much for your indication. Although we did not use video cameras or other standardization, we gave instructions for the hip and knee joint angles to be as follows. We also note in the limitation that we do not standardize angles for video cameras and other devices.

lines 137-142

The NHE [19, 20] was performed in which the participants were instructed to lean forward gradually at the slowest possible speed from a kneeling posture with the knee joint flexed 90° to a prone position with the arms crossed at the chest and the hip joint fully extended. In contrast, for RHC, participants were instructed to change the kneeling position, with their buttocks immediately superior to their ankles and their knees, and hips fully flexed [16].

lines 295-299

Third, previous studies set failure criteria for hip flexion angle and discarded NHE with hip flexion angles higher than 20° [40]. In this study, subjects were verbally instructed not to flex their hip joints during NHE without specifying the hip joint flexion angle. Therefore, future studies should define and examine the hip flexion angle.

10. Line 133-137; A reference is required that this method is valid and reliable.

Response: Thank you very much for your indication. We have added a citation to the reference.

Change: 133-137

The force against the ankle brace in the vertical direction was measured by the load cell connected to a PowerLab16/35 (AD Instruments, Bella Vista, Australia), synchronously transferred the force data to a personal computer (VersaPro; NEC, Tokyo, Japan) at 1000 Hz [6].

11. Line 152-159; Reference should be given for electrode placement.

Response: Thank you very much for your indication. The position of the s-EMG electrode attachment was based on Surface Electromyography for the Non-Invasive Assessment of Muscles. Information on references has been added.

Change: lines 153-154

The position of the s-EMG electrode attachment was based on Surface Electromyography for the Non-Invasive Assessment of Muscles[21].

12. Line 171; Is post-power analysis done?

Response: Thank you very much for your indication. In this study, only the Wilcoxon signed-rank test and the paired t-test were used.

DISCUSSION

13. I would recommend making the discussion section more fluent. Here it is nice to support and compare your results with current literature. However, while the study was about peak ecc force and muscular activations, the discussion was based on morphological features such as fascicle length. More relevant literature on the study subject and hypothesis can be used.

Response: Thank you very much for your indication. Morphological characteristics like fascicle length, which was not measured in this study, dominated the discussion. The content regarding fascicle length has been changed to make the text more relevant to this study. We would appreciate it if you could review the text.

14. Finally, you can define the peak ecc force during NHE and RHC exercises more clearly (for which muscle(s)), which highlights your working hypothesis (which is different from the current literature).

Response: Thank you very much for your indication. We think that the peak muscular force during NHE and RHC exercises is a combination of the strength of several muscles, including the hip, and trunk extensors, gastrocnemius, and hamstring. We have added text stating these things.

Change: lines274-280

Nishida et al.[6] detected no significant relationship between peak torque during NHE and knee flexion torque measured with an isokinetic dynamometer in the supine position. Therefore, the eccentric force measured in this study may reflect the combined strength of various muscles, such as the hip, and trunk extensors, gastrocnemius, and hamstring. The factors related to the eccentric force exerted during NHE and RHC should be investigated.

References

15. It's good that about 50% of the references were published in the last five years.

Response: Thank you very much. We believe it is important for researchers to learn about recent research findings. I will continue to strive and improve in the future.

Reviewer #3

1. In my opinion, the manuscript can be published but I would recommend the authors to modify the figures so that the pictures provided are of high quality and without background that could affect the understanding of it.

Response: Thank you very much for your indication. We have taken the photos again.

Change: Figure 1

2. They should also modify the figure accompanying the results and make it much more visual, either by using a design programme or by modifying it to violin plots, which are much more appropriate.

Response: Thank you very much for your indication. As for figure, we changed it to a graph using violin plots.

 Change: Figure 2

Reviewer #4

1. L74 In this part, you wrote too much about Pollard's experiment. Please sum up this experiment.

Response: Thank you very much for your indication. We have included a more summarized version of Pollard's studies

Change: lines 73-76

Similarly, to compare the training effects of NHE and RHC, Pollard et al.[16], conducted a study in three groups; NHE with body weighted (NHEbodyweight), NHE with weighted (NHEweighted) and RHC with weighted (RHCweighted) in a 6-week training intervention. 

2. L85 I think writing "in addition to the hamstrings" is unnecessary.

Response: We have removed "in addition to the hamstrings" as you suggested. 

Change: lines 80-81

A previous study showed that s-EMG of the GM, Erector Spinae (ES), and gastrocnemius were exerted during NHE [18].

3. L87 Please write the detail about this measurement. It would help if you wrote why the peak eccentric force measurement is essential.

Response: Thank you very much for your indication. The peak eccentric force is considered an indicator of hamstring ECC strength, such as used to predict the risk of HSI. Therefore, we believe peak eccentric force measurement is important. We have added text to convey that peak eccentric force was measured as an indicator of hamstring ECC strength.

Change: lines 83-87

Recently, instruments have been developed to measure the peak eccentric force during NHE and RHC. The peak eccentric force is considered an indicator of hamstring ECC strength, such as used to predict the risk of HSI [19, 20]. Therefore, measuring peak eccentric force in addition to s-EMG may assume the amount of load on each muscle during NHE and RHC.

4. L109 About the exercise habit, this is essential information. For example, it differs depending on whether it is a power or endurance sport. Please write the detail about it.

Response: Thank you very much for your indication. We have described the exercise habits of the participants in more detail.

Change: lines 106-108

This study included the fifteen healthy male university students (age of 20.9 ± 0.9 years, height of 171.1 ± 5.1 cm, and weight of 68.2 ± 9.0 kg) with habitually performed resistance training including that for the knee flexors 2–3 times a week.

5. L133 What is the load cell sensor? Please explain this. Furthermore, write the detail of this equipment; name, company...

Response: Thank you very much for your indication. The phrase "load cell sensor" was incorrect. We have changed the text again and would appreciate your confirmation.

Change: lines 133-137

The force against the ankle brace in the vertical direction was measured by the load cell connected to a PowerLab16/35 (AD Instruments, Bella Vista, Australia), synchronously transferred the force data to a personal computer (VersaPro; NEC, Tokyo, Japan) at 1000 Hz [6].

---

## [Decision Letter · Decision Letter 1]

2 Oct 2023

PONE-D-23-17165R1Eccentric force and electromyogram comparison between the eccentric phase of the Nordic hamstring exercise and the razor hamstring curlPLOS ONE

Dear Dr. Nishida,

Thank you for submitting your manuscript to PLOS ONE. After careful consideration, we feel that it has merit but does not fully meet PLOS ONE’s publication criteria as it currently stands. Therefore, we invite you to submit a revised version of the manuscript that addresses the points raised during the review process.

We look forward to receiving your revised manuscript.

Kind regards,

Esedullah Akaras

Academic Editor

PLOS ONE

Journal Requirements:

Reviewers' comments:

Reviewer's Responses to Questions

**Comments to the Author**

1. If the authors have adequately addressed your comments raised in a previous round of review and you feel that this manuscript is now acceptable for publication, you may indicate that here to bypass the “Comments to the Author” section, enter your conflict of interest statement in the “Confidential to Editor” section, and submit your "Accept" recommendation.

Reviewer #1: All comments have been addressed

Reviewer #4: All comments have been addressed

2. Is the manuscript technically sound, and do the data support the conclusions?

Reviewer #1: Yes

Reviewer #4: Yes

3. Has the statistical analysis been performed appropriately and rigorously? 

Reviewer #1: Yes

Reviewer #4: Yes

4. Have the authors made all data underlying the findings in their manuscript fully available?

Reviewer #1: Yes

Reviewer #4: Yes

5. Is the manuscript presented in an intelligible fashion and written in standard English?

Reviewer #1: Yes

Reviewer #4: Yes

6. Review Comments to the Author

Reviewer #1: In my opinion, the sample size is low to have robust results. It would be interesting to add more data to the study.

As for the rest of the comments, the authors have reviewed them satisfactorily.

Reviewer #4: Please rewrite L88-L91. The conjunction "therefore" has been followed. Please reconsider the connection of these sentences.

7. PLOS authors have the option to publish the peer review history of their article (what does this mean?). If published, this will include your full peer review and any attached files.

Reviewer #1: No

Reviewer #4: No

---

## [Author Response · Author response to Decision Letter 1]

10 Oct 2023

Thank you very much for reviewing our manuscript and providing valuable comments and suggestions that have helped us improve our manuscript. We have responded to your comments below and revised our manuscript accordingly. 

In the revised manuscript, the changes are highlighted in yellow with red letters. 

We hope that you will find our responses and revisions, considering your comments, adequate and that the manuscript has improved. We appreciate your further review of our manuscript and hope we have an opportunity to publish it in this journal.

Reviewer #1

Comment: In my opinion, the sample size is low to have robust results. It would be interesting to add more data to the study. As for the rest of the comments, the authors have reviewed them satisfactorily.

Response: Thank you very much for your indication. We have mentioned this in the limitation, but as you pointed out, we are considering the possibility that the sample size is small. Therefore, we believe it is necessary to conduct large-scale studies with larger sample sizes in the future.

Reviewer #4

Comment: Please rewrite L88-L91. The conjunction "therefore" has been followed. Please reconsider the connection of these sentences. 

Response: Thank you very much for your indication. The conjunction "therefore" was used in succession, which made the sentences less connected. We have revised the text to take this into account and would appreciate it if you could confirm this.

Change: lines 86-89

Measuring peak eccentric force in addition to s-EMG can provide insights into the load on each muscle during NHE and RHC. However, no previous studies have compared peak eccentric force during NHE and RHC.

---

## [Decision Letter · Decision Letter 2]

23 Oct 2023

Eccentric force and electromyogram comparison between the eccentric phase of the Nordic hamstring exercise and the razor hamstring curl

PONE-D-23-17165R2

Dear Dr. Nishida,

We’re pleased to inform you that your manuscript has been judged scientifically suitable for publication and will be formally accepted for publication once it meets all outstanding technical requirements.

Kind regards,

Esedullah Akaras

Academic Editor

PLOS ONE

Additional Editor Comments (optional):

Reviewers' comments:

Reviewer's Responses to Questions

**Comments to the Author**

1. If the authors have adequately addressed your comments raised in a previous round of review and you feel that this manuscript is now acceptable for publication, you may indicate that here to bypass the “Comments to the Author” section, enter your conflict of interest statement in the “Confidential to Editor” section, and submit your "Accept" recommendation.

Reviewer #1: All comments have been addressed

Reviewer #4: All comments have been addressed

2. Is the manuscript technically sound, and do the data support the conclusions?

Reviewer #1: Partly

Reviewer #4: Yes

3. Has the statistical analysis been performed appropriately and rigorously? 

Reviewer #1: Yes

Reviewer #4: Yes

4. Have the authors made all data underlying the findings in their manuscript fully available?

Reviewer #1: Yes

Reviewer #4: Yes

5. Is the manuscript presented in an intelligible fashion and written in standard English?

Reviewer #1: Yes

Reviewer #4: Yes

6. Review Comments to the Author

Reviewer #1: Thank you for the oportunity to review this manuscript. All comments have been addressed by the authros.

Reviewer #4: Thank you for changing the article. The text became clearer and easier for readers to understand. I think it is good.

7. PLOS authors have the option to publish the peer review history of their article (what does this mean?). If published, this will include your full peer review and any attached files.

Reviewer #1: No

Reviewer #4: No

---

## [Editor Report · Acceptance letter]

8 Dec 2023

PONE-D-23-17165R2 

Eccentric force and electromyogram comparison between the eccentric phase of the Nordic hamstring exercise and the razor hamstring curl 

Dear Dr. Nishida:

I'm pleased to inform you that your manuscript has been deemed suitable for publication in PLOS ONE. Congratulations! Your manuscript is now with our production department. 

Kind regards, 

on behalf of

Dr. Esedullah Akaras 

Academic Editor

PLOS ONE